# Development of Bioactive Scaffolds for Orthopedic Applications by Designing Additively Manufactured Titanium Porous Structures: A Critical Review

**DOI:** 10.3390/biomimetics8070546

**Published:** 2023-11-13

**Authors:** Mikhail V. Kiselevskiy, Natalia Yu. Anisimova, Alexei V. Kapustin, Alexander A. Ryzhkin, Daria N. Kuznetsova, Veronika V. Polyakova, Nariman A. Enikeev

**Affiliations:** 1N.N. Blokhin National Medical Research Center of Oncology (N.N. Blokhin NMRCO), Ministry of Health of the Russian Federation, 115478 Moscow, Russia; kisele@inbox.ru; 2Department of Casting Technologies and Artistic Processing of Materials, National University of Science and Technology “MISIS”, 119049 Moscow, Russia; 3Laboratory for Metals and Alloys under Extreme Impacts, Ufa University of Science and Technology, 450076 Ufa, Russiaalex.sandr00@bk.ru (A.A.R.); littlepikelet@gmail.com (D.N.K.); vnurik@gmail.com (V.V.P.); nariman.enikeev@gmail.com (N.A.E.); 4Laboratory for Dynamics and Extreme Characteristics of Promising Nanostructured Materials, Saint Petersburg State University, 199034 St. Petersburg, Russia

**Keywords:** additive manufacturing, bioactive scaffolds, porous materials, finite element simulation, pore design, microstructure, biocompatibility, mechanical properties, titanium alloys

## Abstract

We overview recent findings achieved in the field of model-driven development of additively manufactured porous materials for the development of a new generation of bioactive implants for orthopedic applications. Porous structures produced from biocompatible titanium alloys using selective laser melting can present a promising material to design scaffolds with regulated mechanical properties and with the capacity to be loaded with pharmaceutical products. Adjusting pore geometry, one could control elastic modulus and strength/fatigue properties of the engineered structures to be compatible with bone tissues, thus preventing the stress shield effect when replacing a diseased bone fragment. Adsorption of medicals by internal spaces would make it possible to emit the antibiotic and anti-tumor agents into surrounding tissues. The developed internal porosity and surface roughness can provide the desired vascularization and osteointegration. We critically analyze the recent advances in the field featuring model design approaches, virtual testing of the designed structures, capabilities of additive printing of porous structures, biomedical issues of the engineered scaffolds, and so on. Special attention is paid to highlighting the actual problems in the field and the ways of their solutions.

## 1. Preface

With recent progress in additive manufacturing (AM) technology, the number of publications on developing porous materials for biomedical applications exhibits an avalanche-alike growth, as denoted in a series of very recent extensive reviews [1,2,3,4,5]. The advantages of AM approaches to engineer fine-structured, property-controlled, and custom-designed products of numerous metallic, ceramic, carbon, and plastic materials stipulated the development of porous materials with enhanced mechanical, biocompatible, and bioactive properties. Successful exploring this area demands the realization of multidisciplinary concepts, joining efforts of prominent researchers in the field of AM technology, computer-assisted design, multiscale simulation and machine learning, tissue engineering, microstructural assessment, property characterization, biomedical studies, orthopedic surgery, and so on. Specialists from these different fields might under or overestimate the possible troubles arising from every particular aspect of developing an “ideal” final product. The present review focuses on the analysis of the newest findings, provided by synergetic efforts of different researchers to engineer porous materials using AM, and highlights critical issues to be solved on the way to manufacture the advanced personalized bioactive implants/scaffolds for innovative applications in orthopedic oncology. We overview the benefits and troubles associated with (i) modeling and experimental issues related to AM production of purposefully designed porous structures; (ii) simulation-based approaches to optimize their functional properties; (iii) biomedical and clinical aspects of developing AM-produced porous specimens to provide required bioactive effects: osteoconductivity and osteoinductivity as well as the capability to use scaffolds as platforms for the local delivery of bioactive compounds into an implanted area. As outlined below, the presented review is focused on the issues related to AM using laser or electron beam powder bed fusion (L-PBF/EB-PBF) for the specific purpose of engineering bioactive porous orthopedic scaffolds of Ti-based metallic materials.

## 2. AM-Aided Engineering of “Ideal” Orthopedic Bioactive Implants

Orthopedic implants represent medical devices that facilitate the healing of bone fractures and defects. Healing of bone fractures involves a regenerative process that, in most cases, results in the restoration of the damaged bone. However, 5–10% of fractures lead to delayed healing or retarded bone union, especially in cases of comorbidities such as diabetes [6]. Complex or compromised bone fractures (i.e., fractures larger than a critical size of <4 cm, with severely damaged surrounding tissues), comorbidities, and improper initial treatment increase the risk of delayed healing and non-union [7]. Additional indications requiring bone healing include bone defects resulting from bone tumor resections, infections, or in the context of prosthetics. Damaged joints and degenerative diseases may require arthrodesis, the artificial induction of a joint bridge between two bones, also known as fusion. Arthrodesis is mostly performed on the spinal joints, the wrist, ankle, and foot. All these conditions require the filling of bone defects and bone grafts. Global sales of orthopedic products are estimated at $55.5 billion in 2022 and include all products used for the treatment of fractures, both internal and external: plates, screws, intramedullary nails, pins, wires, staples, and external fixators; spinal implants and instruments [8]. Patients with various bone diseases, particularly with injuries, require reliable bone defect replacement with biocompatible materials. Novel orthopedic solutions also demand the implementation of bioactive effects.

Note that the number of investigations in this multidisciplinary field is skyrocketing with each year, and it naturally entails certain confusion in the interpretation of terms used by different research groups. For example, the definition and understanding of the “bioactive” term is critical for a proper analysis of the numerous available publications, implying quite a broad interpretation of its meaning bearing a pretext for polemics. The author of [9] critically assesses the relevance of using this term in the area of dentistry and reveals controversy in its interpretation throughout many reported studies. A general academic definition of a bioactive compound—“*having or producing an effect on living tissue*”—allows defining all materials used in internal implants as bioactive since the reaction of living tissues to any introduced substance is inevitable. Another common definition, also used in ISO 23317, severely restricts the meaning of “bioactivity” to the “*property that elicits a specific biological response at the interface of the material*, *which results in the formation of a bond between tissue and material*” [9,10]. Apart from that, many studies refer to bioactivity as an ability to emit chemicals having antibacterial, cytotoxic, or growth-assisting effects, which can also be reasonable within the general concept. For example, the International Union of Pure and Applied Chemistry (IUPAC) defines “bioactivity” as a “*qualifier for a substance which provokes any response from a living system*” as opposed to properties of bioinert products (such as stents or contact lenses) which are designed to cause the least effect on the living tissues [11]. Bioactive implants, from this point of view, should provide a biological response at the implantation site—whatever it is related to better osteointegration as well as tissue regeneration, local healing, suppression of the immune response, infection prevention, and so on. This review does not pretend to judge which definition should be accepted and used by all the researchers; however, it is important to note that we keep the latter interpretation when considering “bioactive implants/scaffolds” in the present review, which is consistent with the conclusion of [9] stating that the implants can be labeled as bioactive if they represent “*vehicles for any of the many biologically-active substances that offer the possibility of true reparative or regenerative responses*” having also a positive therapeutic effect.

The demand of orthopedic oncology for bioactive implants as carriers of pharmaceuticals is fueled by the prevalence of bone sarcoma, characterized by high morbidity and mortality, especially for children and adolescents [12]. The most common types are osteosarcoma (56%), chondrosarcoma (10%), and Ewing’s sarcoma (34–36%) [13]. Despite the advancements in modern oncology, systemic chemotherapy remains the most used method for treating malignant tumors. However, when administered intravenously, only a small portion of the anti-tumor cytostatic agents reach the tumor, affecting normal tissues and leading to undesirable side effects. Local chemotherapy via specially loaded implanted devices is a highly promising method for enhancing the effectiveness of anti-tumor treatment and minimizing systemic collateral effects [14]. The implants could also be used as platforms to emit antibiotics or antibacterial compounds (such as Ag nanoparticles), favoring local antimicrobial efficacy [15]. Thus, advanced implants could be considered not only as bioinert orthopedic devices but also as drug delivery systems for local chemotherapy/immunotherapy as well as for the promotion of antibacterial effects.

Another vitally important ability of the implants relates to their interaction with living tissues to provide desired vascularization, osseointegration, and osteoinduction [16]. Successful ossification and vasculature development require proper permeability and fine-tuning of the interplay of complex compounds with different bioactive properties (growth factor, functional molecules, mesenchymal stem cells, and so on). Implant porosity presents room for infusing various functional agents; at the same time, the trapped air spacings would be filled, thus preventing prospects of air embolism [16].

Orthopedic implants can be manufactured of different organic and inorganic compounds featuring, among many, such materials as poly(lactide-co-glycolide), poly(ε-caprolactone), Cervi Cornus Colla, poly(ethylene glycol) maleate citrate, polylactide, poly(methyl methacrylate), peptide-based hydrogel, hyaluronic acid, collagen, demineralized bone matrix, xenogeneic antigen-extracted cancellous bone, various polymers and so on as well as calcium phosphates, hydroxyapatites, ceramics, bioglasses, and metallic materials [17]. These options provide versatile choices of combining various mechanical, topological, chemical, and biological characteristics to design and engineer advanced orthopedic systems, which can also be categorized as biodegradable or permanent implants.

Thanks to their excellent strength, fatigue, and wear properties, biocompatible metallic materials are applied when load-bearing capacity and fixation or replacement of large bone fragments are required. Recent reports consider bioresorbable metals (such as Mg-or Zn-based alloys) as candidates for engineering an “ideal” scaffold [18] with a controllable corrosion rate, which would make it dissolved with the rate corresponding to the bone regeneration after imposing the required repairing, regenerative and therapeutic effects [19]. However, these alloys still exhibit a faster degradation rate, provoking premature failure of manufactured implants [18]. Another problem is related to difficulties in the preparation of powders (characterized by high flammability, oxidation, and evaporation [20]), leading to serious troubles with AM-producing defect-free articles of Mg-based alloys. Despite the efforts taken to control the corrosion rate and optimize alloys’ composition and AM production of Mg-based materials [21], biodegradable metallic scaffolds are still challenging to start a long road to clinical application [20].

Biocompatible metals and alloys (such as Ti, Ta, Co-Cr, etc.) represent the most popular materials widely used to manufacture permanent orthopedic devices. However, these conventionally produced materials have higher elastic moduli (100–140 GPa for Ti alloys [22], 210–253 GPa for Co alloys [23], and 190–210 GPa for stainless steel [24]) than that of bone (0.5–20 GPa) [25]. Such an incompatibility of stiffness between an implant and a bone can result in the stress shielding effect [26], leading to implant loosening or autogenous bone fracture. Most fully dense metallic implant materials used in clinical practice also suffer from problems such as implantation-induced infection, unstable interface with host tissue, biomechanical mismatch, and limited service life. Due to these clinical shortcomings, current research is focused on developing novel devices that can prevent postoperative infection, provide early stability and rapid healing, and enable the remodeling of surrounding tissue structures [27].

As a result, in recent years, there has been great interest in engineering metallic materials with a porous structure, which allows for purposeful tuning of their mechanical and functional performance. By varying the size and fraction of pores, it is possible to design a metallic implant with optimal mechanical properties and compatibility with bone tissue, which can prevent osteonecrosis and disruption of osteointegration [1,2,3,4,5,17]. In addition, the internal porous structure promotes adhesion, proliferation, and differentiation of mesenchymal stem cells in the osteogenic direction. The porous structure also provides a larger surface area for vascularization and osteointegration, promoting biological fixation of the implant and bone tissues [5,16].

AM technologies open cardinally new prospects to design metallic porous structures for advanced personalized bioactive orthopedic implants of the next generation [1,2,3,4,5,16,17]. The unprecedented capabilities of AM allow the printing of multifunctional cellular implants with a thoughtfully designed, highly complicated internal topology that cannot be reproduced using traditional approaches. AM techniques are capable of tailoring metallic implants to be most compatible with the replaced bone fragments by mechanical, chemical, and biological properties. Among all, Ti-based alloys, in particular, Ti-6Al-4V alloy, attract the utmost attention of researchers as a clinically approved material for developing AM-driven orthopedic solutions thanks to the unique combination of functional properties including excellent biocompatibility, high fracture toughness, good wear, and corrosion resistance [22,28], while pore engineering allows adjusting stiffness to prevent stress shielding effect [29]. Recent advances in the optimization of AT processes to design a topology of Ti porous structures allowed us to find promising paths to fit multiple demands for desired combinations of mechanical and biological parameters of implants, mimicking properties of bone tissue and ensuring successful bone regeneration and tissue integration [30]. Tunable pore architecture also enables enhancing permeability and, hence, better osteointegration [31,32], as well as providing a space to be filled with various bioactive components [16].

Thus, the AM technology appears as the Holy Grail to tailor a new generation of “ideal” multi-functional Ti-based bioactive scaffolds [33]. However, despite tremendous activity in the field, there are still many troublesome issues in the design and production of reliable, high-performance orthopedic devices of AM porous materials.

A typical strategy of designing a product starts from the definition of desired properties for an ideal implant [18] comprising the requirements for mechanical and biological performance. Then, tasks are defined, comprising:(i)Pore design with feedback from (i1) experimental producing of the designed structure; (i2) simulation to estimate strength, anisotropy, stiffness, and fatigue resistance depending on the pore geometry; (i3) experimental validation of numerical results;(ii)Appropriate choice and thoughtful adjustment of 3D printing equipment, powders, and processing parameters depending on the pore design and product configuration;(iii)Account for AM-induced defects and post-treatment procedures such as annealing, etching, loading with bioactive compounds, sterilization, trapped air and powder removal, and so on.

Note that the desired properties of the “ideal implant”, in turn, impose specific, often contradictory, demands for the topological composition of the porous material. For example, larger pores would enhance permeability and, hence, promote vascularization, while smaller pores facilitate osteoblast cell colonization [34], see Figure 1.

Again, enlarging pores allows for reduced stiffness and, at the same time, decreasing mechanical strength [5]—and so on. As a result, the task of developing an “ideal” pore architecture involves multidimensional optimization of numerous arrays of parameters, which are often independent of each other—like stiffness and cell proliferation. This dimensionality goes to infinity, assuming a multiplicity of different types of pore geometries, implants, AM features, materials, coatings, bioactive components, their compositions, and so on. Due to these reasons, there is still limited success in using AM for the development of advanced biomedical products in clinical practice.

Assuming the issues above, let us limit the area of our focus to a specific case related to the design and AM production of permanent porous titanium bioactive scaffolds for orthopedic oncology. Solving the related problems requires a deep interdisciplinary understanding of the structure-topology-property interrelationships from physical, chemical, computational, and biomedical points of view. There is often a gap between different field scientists in understanding issues critical for the engineering, producing, and biomedical aspects here. We are trying to present an interdisciplinary point of view, coordinated among the specialists from several sides. In the next section, we overview achievements and troubles at each stage of product development: computer-assisted design of pore geometry, AM producing of the designed porous structures, and computer simulation of mechanical properties with experimental validation.

## 3. AM Approaches to Print Porous Structures

Modern 3D-printing technologies offer an opportunity to produce a wide range of materials with pre-defined internal geometries, dimensions, and AM-induced microstructures. This subsection briefly focuses on the AM solutions suitable for the manufacturing of fine porous titanium alloys approved to be used in articles for medical applications.

### 3.1. AM Techniques to Print Porous Biocompatible Products of Ti Alloys and the Features of As-Printed Materials

3D printing is known to introduce both internal defects and geometrical deviations in the manufactured workpieces caused by the quality of powders, non-optimal processing parameters, and native peculiarities of AM technology [35]. These features may affect the functional performance of the printed articles. Achieving high-quality products requires laborious monitoring procedures involving instrumental control and computer-assisted engineering, including machine learning [36]. Adopting AM for medical products is impossible without the implementation of specialized approaches allowing to highly precisely print defect-free, thoughtfully architectured articles.

Modern AM techniques can provide rapid prototyping of complicated 3D structures via progressive layer-by-layer joining materials programmed with the model data [37]. There are numerous AM methods, each with its own ups and downs depending on the specific application, as denoted in many comprehensive reviews [38,39,40]. A detailed overview of numerous developed AM methods and approaches is beyond the scope of this paper. It is important to note that high-precision printing of fine structures with complicated geometry is possible using AM methods based on the fusion of ultra-dispersed powders powered by laser or electron beam, which have become the most popular techniques to manufacture medical devices [1,2,3,4,5,40]. The former is often referred to as the Selective Laser Melting (SLM) technique. However, SLM represents rather a proprietary name of the AM process owned by the inventor, SLM Solutions Group AG, Lübeck, Germany. To keep the consistency, both techniques are categorized using the main prototyping principle and labeled as laser or electron beam powder bed fusion (L-PBF and EB-PBF, respectively), as we will refer to them hereinafter.

PBF techniques enable the printing of porous structures with the finest available dimensions of pores (starting from 20–25 µm) [41] of high-quality biocompatible alloys. Even if modern studies predict an optimal pore size to be in the range of 300–600 µm (see below), the AM process must ensure much higher precision of printing than the designed pore dimensions. Even to compose 300 µm-sized pores, one needs to provide a printer resolution ability to reproduce geometrical features of several tens of microns. PBF techniques are capable of producing such precise porous structures with smooth surfaces [42,43,44].

Printing parameters for porous structures can significantly vary depending on the specific material and application [45]. Printing percolating porous structures with sophisticated fine-scale internal geometries demands the highest achievable precision [46] provided by using powders with the least possible particle size as well as the smallest laser beam size with corresponding accuracy of the positioning system accompanied with appropriate laser power/energy density [47,48,49]. AM of bulk high-precision products should also consider support designs of a printed article and its orientation [50].

PBF processing parameters can have a significant effect on the homogeneity of the structure and properties of printed samples, as well as provoke the formation of undesired AM-induced defects such as voids, cracks, and unmelted particles [51,52]. The location of the product on the printing platform can entail scattering in properties of the printed objects with deviations varied from 5 to 17% depending on their location [52]. High residual stresses may arise due to heterogeneous heating and rapid solidification during the PBF process, while post-printing annealing can affect the mechanical performance and phase composition of the printed articles as well as lead to the formation of cracks, shape distortion, and detachment from the supports [53,54]. As a result, the production of defect-free AM metallic products requires laborious preliminary equipment adjustment and specific post-processing [55].

The microstructure of the AM materials is usually characterized by a fine-grained structure that arises from the rapid solidification of melted metal powders. The size and morphology of the grains and fragments can vary depending on the alloy’s composition, AM parameters, and heat transfer during the process [45,48,49,51]. Phase composition can also be affected by the printing processes of Ti alloys accompanied by optional precipitation of intermetallics [56]. At high cooling rates, depending on the class of a Ti alloy, a structure from a quasi-equilibrium Widmanstett structure consisting of α-phase plate packets [57] to a non-equilibrium fine-dispersed acicular martensite structure [58] with a high density of dislocations and twins [59] can be formed out of β-phase grains. The formation of such non-equilibrium structures leads to a significant increase in the strength and a loss in ductility of printed Ti alloys compared to those produced by traditional methods of metal forming [60,61]. In order to compensate for the effect of AM-induced microstructural features, specified heat treatments in the form of tempering or aging are applied [62,63].

The aforementioned factors can influence the mechanical properties of AM-produced materials. In addition, PBF-printed specimens can exhibit a specific crystallographic texture with the preferential alignment of grains along the construction direction, which introduces considerable anisotropy of mechanical properties, especially in materials with low symmetry, such as Ti alloys [64].

The PBF 3D printing process can additionally result in heterogeneous surface roughness due to the layered deposition of material, which is important from a biomedical point of view. To improve surface quality, dedicated post-processing methods such as chemical or electrolytic etching should be used [65]. The texture and topography of the surface can affect the wettability, adhesion, and corrosion resistance of printed materials [66].

Additional trouble with the printed porous structures is related to the intrinsic feature of AM technology: unmelted particles can be trapped in the internal cavities of the printed product [67]. The presence of trapped powders is undesired for articles designed for medical applications, and cleaning by powder recovery systems is required. Standard techniques such as air jet cleaning might be insufficient to release trapped powder from porous specimens with complicated internal geometry [67], and chemical or ultrasound vibration procedures must be applied. In turn, these additional treatments might change the primarily designed cellular geometry, which has to be accounted for during their design.

The issues above can significantly impact the mechanical properties and functional characteristics of printed components. Therefore, it is crucial to optimize process parameters, including laser power, scanning speed, and powder feed rate, to minimize the number of defects and ensure material integrity [68]. Specific features of the AM alloys, such as refined microstructure, increased defect density, changed phase composition, and AM-induced texture, are very important to be taken into consideration in computer-aided design of the porous structures.

### 3.2. Computational Techniques for AM-Aimed Cell Design and Virtual Testing of Porous Structures

Bioactive scaffolds need a purposeful design of porous structures with fine, sophisticated geometry to satisfy numerous requirements of regulated mechanical properties, osteo-inductivity and conductivity, biocompatibility, permeability, and capacity for drug loading.

Understanding how the design of porous structures defines the properties of printed articles is highly important to achieving high-quality medical products with desired performance. The models of porous structures are characterized by the spatial arrangement of cells consisting of pores and inter-pore walls, the pore size and their distribution, cell geometry, configuration of inter-pore partitions, the type of pore relief, etc. [69]. The mechanical and physical properties of porous structures explicitly depend on the topology of the cells. The analysis of the relationship between structure and properties shows that the mechanical and physical properties of porous materials depend not only on the chemical composition of the material but also on the geometric characteristics of the elementary cell or cell blocks forming the porous/cellular material.

At the same time, it should be noted that for the normal development of bone tissue, porous materials must provide the diffusion of fluids and nutrients, as well as the removal of metabolic waste [70,71,72]. It should also be considered that the structure of the material is important for the functioning of bone tissue during and after the process of regeneration and remodeling [73,74,75]. This subsection considers computer-aided design of the pore geometries and numerical approaches to simulate the target properties of the cellular structures.

There are various computer-aided methods to design porous materials regarding the configuration of their internal topology. They can roughly be classified as constructed with (i) spatially arranged cells composed of struts, (ii) triply periodic minimal surfaces (TPMS), and (iii) irregular bio-inspired stochastic or Voronoi tessellation structures [5,76,77,78,79,80]. The latter two techniques provide versatile capabilities to engineer porous scaffolds with controllable mechanical performance and enhanced cell colonization and proliferation [78,81]. While irregular structures mimic the natural composition of bone tissues, their design, and AM processing are laborious because of higher scattering of results among the generated structures as well as poorer basis for comparison of their performances.

Let us consider the basic principles for the computer-aided development of porous structures on the example of the most popular approach for the flexible design of versatile porous materials based on the mathematical representation of cellular structures by TPMS [81,82]. This approach provides an easy-to-implement yet powerful tool to mimic the topological, mechanical, physical, and biological properties of natural bone [82]. This architecture also enables the storage of large volumes of liquids, such as medications and columnar cells, in a small volume of porous material. [83].

TPMS is an infinite and periodic curved surface that does not contain self-intersecting fragments and allows for the creation of homogeneous structures. These surfaces have crystallographic group symmetries: cubic, tetragonal, hexagonal, rhombic. TPMS is formed using an implicit method, i.e., using unambiguous functions of three variables, and the surface is defined using three axis parameters [84] (*x*, *y*, *z*). An example of describing a TPMS surface is an equation of the general type (1):cos *αx* + cos *βy* + cos *γz* = *c*,(1)
where *α*, *β*, *γ* are parameters that determine the sizes of the base cell along the *x*, *y*, and *z* axes, and the constant *c* determines the density of the structure. That is, this equation represents a set of trigonometric functions that together satisfy the equality ϕ(*x*, *y*, *z*) = *c*, and this function ϕ(*x*, *y*, *z*) is an isosurface evaluated by the isovalue *c*. Variable density, cell size gradients, hybridization, hierarchy, etc., are achieved by controlling the implicit function (cos(*x*, *y*, *z*) or sin(*x*, *y*, *z*)) and the constants [85].

Currently, the most widely used programs for creating porous structures include Triangulatica, nTopology, Gen3D Sulis, Autodesk Fusion 360, Netfabb, etc., as well as the free MSLattice plugin [86], which was used to create models in MATLAB (R2022b), SpaceClaim (for ANSYS R19.1), and OpenSCAD (2021.01). Figure 2 illustrates the opportunities to construct porous cylinders differently designed based on various models.

Parameters for the created TPMS models are provided in Table 1. Models (1–4) represent the possibility of varying the cell wall/pore size ratio within one design (IWP). Models (5–8) display a variety of designs represented using diamond, gyroid, and strut-based octa-alike structures.

Important to highlight a problem related to the mismatch between the designed and printed models, which is originated from the 3D printing process listed above: heterogeneous solidification of melted powders; residual stresses, which can cause noticeable deviation of the product’s geometry from the desired shape; inaccuracies associated with the finite size of the laser beam, comparable to the size of the printed pores and non-perfect positioning [1]. Figure 3 [87] shows a comparison of the designed and printed fragments of the cellular structure, showing AM-induced inaccuracy in the reproduction of the desired cell walls. Note that the authors of [87] used very fine powders with volume-weighted equivalent diameters as small as *d*_10_ = 12.1 µm, *d*_50_ = 23.6 µm and *d_9_*_0_ = 37.6 µm. A laser beam size was 30 µm to reliably print the cells with the minimal strut thickness of 100 µm. Figure 3 indicates that struts of the printed cellular structures had considerably appeared roughness as compared to the designed CAD models. This inconsistency can considerably affect the performance of the porous scaffolds and has to be considered during the manufacturing of the scaffolds.

### 3.3. Virtual Optimization of Porous Structures for Biomedical Applications

Assuming that the studies on developing pore geometry for superior functional performance of bioactive scaffolds require optimization of the design and service properties in multi-dimensional parameter space, the application of computer simulation seems to be vital for further progress in the area. Finite-element modeling (FEM), dedicated to virtual testing of differently designed porous structures and accompanied by experimental validation, represents the most overwhelmingly growing field of study in developing bioactive implants. Numerous reviews testify that virtual testing of porous articles is able to adjust key parameters of their mechanical performance demanded by biomedical applications, such as strength, stiffness, and fatigue resistance, as well as biomedical properties [1,2,3,4,5,88]. However, high-precision computer simulations of the sophisticated internal geometry of the developed products require substantial computation power. Moreover, the mechanical response of generic porous configurations may not be calculated in full-field approximation, even using a representative volume element approach at reasonable computational cost [89]. The limitations imposed by meshing allow us to numerically study the materials with low porosity only (less than ~20%) without pore interconnections [90]. Mean-field approximations reducing demands for computational power do not guarantee proper accuracy if complicated cases of sophisticated microstructures or non-linear behavior are considered [89,91]. Multi-scale or multi-process simulations (such as those considering fluid dynamics of drug release by porous structures) require immense computation power available in supercomputers with parallel implementations of computational tools [92]. Another major challenge for computer simulation of porous implants is presented by a problem of interrelation between their topology and biomedical properties: cell proliferation and colonization, vascularization, osteointegration, and osteoinduction, which often impose contradictory requirements to the pore geometry and the internal surfaces. This issue seems to be numerically unresolvable, requiring in-depth, extensive experimental in vitro and in vivo studies.

Modern computational solutions based on machine learning could significantly contribute to solving these problems. Several studies demonstrate a promising potential of artificial intelligence (AI) to optimize and predict a number of mechanical parameters such as compressive strength, tensile strength, shear, and Young’s modulus [93,94]; to evaluate stress-shielding effect [95]; to design drug delivery systems [96]; to account for printing quality [97]; to mimic natural cellular and porous structures [98]. Very few studies report on the first encouraging attempts to link geometric and mechanical parameters of scaffolds with their physical and chemical properties correlated with in vitro and in vivo biocompatibility tests on cell-material interaction [99,100].

However, AI-based approaches have significant limitations imposed by training procedures, which require vast numbers of carefully selected datasets and are accompanied by interpolation and black-box problems [93]. These issues may considerably obstruct advancing AI solutions in tissue engineering despite the great anticipation of benefits to be provided by neural networks. The reliable design of an ideal porous scaffold with superior mechanical and biomedical performance still needs extensive systematic studies and collection categorized by numerous computed and experimental datasets.

So far, the most developed approaches for applied research in the field of engineering are still related to FEM-based approaches providing reasonable deviations between predicted and validated printed structures and their mechanical properties [101,102], including failure prediction and fracture patterns in different deformation modes [103]. These approaches can result in developing digital twin technologies [104]; it is also important to involve multiscale simulations to account for microstructure and crystallographic texture effects which can be an issue for HCP materials [105]. However, prediction of the clinical performance of developed scaffolds is still a major challenge for reliable numerical predictive solutions.

## 4. Porous Scaffolds for Bone Tissue Engineering: Biomedical Issues

Among the modern AM technologies emerging for bone tissue engineering, bioprinting methods are often declared as allowing for the production of biomimetic matrices. Bioprinting is based on the precise deposition of a composite biomaterial, consisting of a combination of hydrogels, cells, and, in some cases, growth factors [106]. To stimulate bone tissue regeneration in bioprinting, an additional cellular component is used: mesenchymal stem cells (MSCs), umbilical cord stem cells, and endothelial stem cell precursors, which are used for neovascularization purposes. However, despite their undeniable advantages, biomimetic degradable matrices have significant drawbacks due to their insufficient mechanical properties, and the gradual biodegradation of the implant can lead to implant failure. Although intensive work in this field continues, the research is still not at the stage of clinical trials. Developing Ti-based porous scaffolds as bioactive implants seem to be more promising from the viewpoint of immediate applied rerearch leading to their implementation to the clinical practice. Here, it is important to consider main biomedical issues related to the subject of this overview.

### 4.1. Porous Matrices

As denoted in the chapters hereinabove, AM methods are widely used to obtain porous metallic structures, allowing for the production of materials with controlled microarchitecture. Porous biological metallic matrices built using the PBF methods have shown promising results in both in vitro and in vivo studies [107]. Porous metal scaffolds are already being used in orthopedics for the implantation of artificial joints and for the reconstruction of bone defects caused by infection, trauma, or tumor resection [108]. The porous structure can reduce risks associated with the stress shielding effect by matching the mechanical properties of the bone and promoting osteointegration in the bone-implant contact zone, providing the transport of nutrients necessary for the viability and differentiation of precursor osteocyte cells. Unlike ceramics and polymers, porous metallic materials have the advantage of balanced mechanical properties and a unique skeletal structure, which expands their application possibilities in orthopedics [109]. Metallic matrices can have a homogeneous or irregular pore size [110,111]. Homogeneous pore size allows for controlled porosity, providing predictable mechanical properties and scaffold biocompatibility [112,113]. However, the human trabecular bone does not have a consistent porosity, so homogeneous porous matrices are not optimal for cell adhesion and proliferation. On the contrary, irregular porous structures, similar to the spongy structure of bone, enhance the biocompatibility of porous matrices and are more favorable for cell growth [27,114,115]. Most non-uniform porous matrices were obtained using the reverse engineering method based on CT imaging, which allows for the simulation of the microarchitecture of natural bone [116]. The mathematical modeling method based on Voronoi-Tessellation enables the construction of approximate models of biomimetic heterogeneous porous materials [117,118]. Methods based on Voronoi-Tessellation not only optimize the microarchitecture of the matrices but also regulate the mechanical properties (elastic modulus and compressive strength) of the porous matrices, which is important for bone tissue engineering [119]. Several research groups have used the Voronoi-Tessellation method to design biomimetic porous matrices. In particular, Fantini et al. [120] created fully connected porous matrices with a trabecular structure and individual geometry. Gómez et al. [121] modeled the trabecular bone structure based on Voronoi-Tessellation using micro-CT in accordance with the key histomorphometric properties of trabecular bone. However, the authors did not provide data on the mechanical characteristics and biocompatibility of these Ti-based trabecular scaffolds. Wang et al. developed approaches to create porous titanium matrices with irregularity in the porous structure based on Voronoi-Tessellation, reproducing the natural trabecular bone. The method allowed for the production of matrices with a gradient distribution of porosity ranging from 60% to 95% and pore sizes from 200 µm to 1200 µm [119]. In vitro studies of the titanium porous matrices showed that the fabricated PBF irregular Ti-6Al-4V matrices based on Voronoi-Tessellation exhibited good cytocompatibility and could be considered promising for orthopedic applications. It has been found that pore characteristics and surface properties significantly influence cell adhesion, proliferation, and differentiation on matrices [112], and cell colonization of matrices is associated with local permeability, which depends on specific surface area and porosity [122].

### 4.2. Cell Geometry

Early studies of materials with different pore sizes have shown that the optimal pore radius for bone ingrowth is 50 μm and can reach up to 150 μm [123,124,125]. According to Lu et al., human osteoblasts can penetrate, colonize, and proliferate inside macro-pores, with a favorable size of over 40 μm [126]. Later, Itala et al. investigated laser-perforated titanium matrices with pore sizes of 50 μm, 75 μm, 100 μm, and 125 μm and found the formation of osteonal structures even in the smallest openings, leading to the conclusion that pore size within this range does not affect bone ingrowth in perforated titanium matrices [127]. Similar results have been obtained in several other studies, considering the minimal matrix pore size to be within the range of 50–100 μm [128,129]. Moreover, authors of [129] mention that osseointegration occurred even in microporosities of about 10 μm, while reducing pore size below submicrometer scale inhibits bone ingrowth. Xue et al. investigated the influence of the pore size of porous titanium on cell penetration and bone ingrowth. The results showed that porous scaffolds with a pore size of 188 μm were covered with cells, but there was a disruption in oxygen and nutrient exchange, leading to cell death within the matrix, and the optimal pore size was found to be over 200 μm [130]. Knychala et al. [131] implanted hydroxyapatite (HA) matrices with the same pore size (500–600 μm) but different strut sizes (100, 120, 150, and 200 μm) and HA matrices with the same interconnection size (120 μm), but different pore sizes (400–500 μm, 500–600 μm, and 600–700 μm), into distal defects of rabbit femoral condyles. The authors obtained ambiguous results, showing that the volume of new bone increased proportionally to the interconnection size. However, significant differences between groups based on the interconnection size were only observed at week 24. The pore size did not significantly affect osteoid matrix formation, except during the first 4 weeks, when greater new bone formation was observed in matrices with smaller pore sizes. At the same time, the authors found that a larger interconnection size contributes to new bone formation and recommended a minimum interconnection size of 120 μm [132]. On the other hand, Shor et al. showed that smaller pores (450 μm) had lower permeability compared to matrices with larger pores (750 μm), which allowed for better penetration of cell suspension into the matrix and cell adhesion [133]. Matrices with larger pores or higher porosity promoted cell viability and proliferation by preventing pore clogging and facilitating better penetration of nutrients and oxygen [134]. Although by now it has been recognized that larger pores in matrices of various origins contribute to better bone regeneration and revascularization, there are several studies indicating a limited role of pore size in osteointegration when using matrices with pore sizes ranging from 350 to 800 μm and porosity from 30% to 70% [135,136]. Based on conducted research, a critical pore size of 200 μm was determined, below which osteoblasts bridged the pore surface without any growth in the pores [130]. Fukuda et al. [137] studied osteoinduction of PBF Ti implants with a canal structure and observed pronounced osteoinduction with pore sizes of 500 and 600 μm. Wauthle et al. [138] investigated PBF tetrahedral porous Ta implants with an average pore size of 500 μm and 80% porosity, finding good in vivo biocompatibility of the implants. Wally et al. [139] analyzed the role of pore size in the porous structure of PBF Ti6Al4V but were unable to draw a definitive conclusion due to the lack of correlation between pore structure and osseointegration.

Taniguchi et al. [30] reported that the PBF porous Ti6Al4V implant with a porosity of 65% and a pore size of 600 μm had comparable mechanical strength to bone, higher fixation capability, and greater bone ingrowth compared to implants with pore sizes of 300 and 900 μm. This is consistent with the recent results by Liu et al. [140] showing that the best osteogenic properties and desired mechanical performance were demonstrated by trabecular bone scaffolds characterized by 65% porosity with a pore size of 550 μm. Wieding et al. [141] concluded that a porous Ti6Al4V matrix with a pore size of 700 μm stabilized segmental bone defects in sheep tarsal bones. Li et al. [142] conducted in vitro experiments to investigate matrices with pore sizes of 500 μm, 600 μm, and 700 μm and porosities of 60% and 70%. Matrix with a size of 500 μm and a porosity of 60% demonstrated superior cell proliferation and osteogenic differentiation of rat bone marrow mesenchymal stem cells (MSCs) in vitro and bone ingrowth in vivo [143]. Liang et al. [108] studied titanium PBF implants with a porosity of 60–70%, which matched the mechanical characteristics of trabecular bone. This study found that a Ti matrix with a porosity of 70% and pore sizes of 313 μm and 390 μm promoted cell proliferation and bone ingrowth. It is hypothesized that not only the size but also the combination of small and large pores plays an important role in cell colonization and proliferation within the matrix. In support of this idea, it has been found that trabecular-like porous scaffolds with full irregularity and higher porosity promote the proliferation and differentiation of osteoblasts due to the combination of small and large pores of various shapes (0–1800 μm) and roughness of 0.25 and 0.5 [108] While increasing the pore size of matrices is believed to enhance osteointegration, the pore size has a limited influence on bone ingrowth in later stages.

Some studies suggest that pores larger than 1 mm in diameter may promote the formation of fibrous tissue [125]. Despite a number of studies, the optimal porosity and pore size of bone ingrowth implants, especially porous PBF implants, are still unclear. Systematic investigation of the impact of porosity and pore size of porous PBF frameworks on mechanical and biological properties is crucial to enhance the reliability and safety of porous PBF frameworks for medical purposes. It is evident that, in addition to pore size, their geometry is also important for osteoinduction. Specifically, structures such as diamond and rhombic dodecahedron are optimal for elastic modulus and provide osteogenic metal matrices. Therefore, a porous Ti6Al4V framework with a rhombic dodecahedron as its elementary cell has the highest mechanical strength and moderate osteogenic properties, while a tantalum matrix with a diamond unit cell structure exhibits excellent osteogenic effects and moderate mechanical strength [144]. However, Lee et al. [145], while studying Ti-6Al-4V samples with pores of different shapes (round, triangular, and rectangular), concluded that the determining factor is not the shape but the pore topography. Therefore, it is necessary to evaluate the curvature and roughness of the surface, as samples with different pore shapes may have different surface topographies.

The analysis conducted shows that despite many studies carried out, optimal dimensions and microarchitecture of matrix pores for bone tissue defect replacement have not yet been determined. New systematic multidisciplinary studies are required to develop the structures comparable to native bone in terms of mechanical properties, pore size, and geometry, providing adhesion, proliferation, and differentiation of osteoblast precursor cells.

### 4.3. Biocoatings of Porous Structures

Orthobiologicals are biological substances such as bioactive molecules, stem cells, or demineralized bone grafts that are used to heal bone defects more quickly. Porous matrices made of titanium alloy, printed on a 3D printer, enhance angiogenesis, osteoblast adhesion, and promote osseointegration. However, titanium alloys are biologically inert, making the attachment between the implant and bone tissue weak. Therefore, surface treatment and implant structure must be considered in order to develop optimal porous implants. Cell differentiation and bone ingrowth are accelerated when the implant surface is covered with a bioactive material or when chemical and thermal treatments are applied, transforming the smooth titanium surface into a rough bioactive surface [146]. It has been demonstrated that chemical and thermal treatment, by immersion in a 5M aqueous solution of NaOH at 60 °C for 24 h, enhances the osteoinductive properties of porous titanium implants and does not require additional use of osteogenic cells or bone morphogenetic protein. Thus, bioactive porous titanium could be an attractive alternative to existing orthopedic implants under load conditions [147]. There are several methods to enhance the biological activity of metallic implants through surface treatment with bioinert metals and simulated body fluid (SBF), which mimics the composition of human plasma. As a result, a biomimetic apatite coating can form on the material surface. One of them is plasma spraying of calcium phosphate, which is one of the most studied methods, and its effectiveness has been confirmed [148]. Another method is a biomimetic coating, where a bone-like apatite layer is created by immersing the metallic matrix in simulated body fluid (SBF) (Hanks’s solution) [149,150]. Kon et al. applied dual-doped hydroxyapatite (Ce^4+^/Si^4+^ doped HAP) coating using centrifugation with extreme centrifugal force, which showed excellent biocompatibility with osteoblast cell line and antibacterial activity [151]. Like the embryonic development of bone, the healing of fractures is directly regulated by key cytokines such as bone morphogenetic proteins (BMPs), transforming growth factor-beta (TGF-β), FGF, parathyroid hormone (PTH), platelet-derived growth factors (PDGF), and the insulin-like growth factor 1 (IGF-1) family [152]. During the cascade of fracture healing, TGF-β and BMP are secreted, facilitating the recruitment of precursor cells, while FGF, PDGF, and IGF induce proliferation, and cellular differentiation is largely regulated by BMP [153,154,155]. As one of the main factors promoting bone tissue regeneration and approved for clinical use, BMP-2 has been utilized in clinical practice for the treatment of spinal fusion. BMP-2 is currently administered locally in the form of a soaked collagen sponge or allografts [156,157]. It has also been shown that systemic administration of recombinant BMP-2, BMP-6, or BMP-7 contributes to bone mass restoration [158,159,160]. Preclinical and clinical studies have shown that local application of recombinant human bone morphogenetic protein-2 (rhBMP-2) can promote bone tissue restoration in cases of bone defects, non-union fractures, spinal fusion, etc. [161,162]. An optimally balanced osteointegrative effect was observed with a concentration of BMP-2 at a dose of 100 μg/g coating. Bone formation during the first 3 weeks was moderate but still higher than in the control, and the process remained at a higher level for 3–6 weeks compared to lower drug concentrations [163]. The osteointegrative activity of BMP-2 is often overshadowed by severe adverse effects that can significantly impair the health and function of the patient’s musculoskeletal system [164,165]. These include ectopic bone formation, paralysis, and neurological disorders [166,167]. It is known that many growth factors, including BMP-2, act pleiotropically. The initiation of a specific polarized effect largely depends on the concentration of the bioactive substance. However, the induced response can be easily reversed using secondary dosage adjustment. BMP-2 acts osteoinductive at low concentrations (from ng to μg) [168] and osteolytic at high concentrations in the mg range [169]. Various methods of applying BMP to the surface of metallic implants have been proposed to enhance their osteoinductivity. Lin et al. Used bioactive peptides isolated from mussels, including adhesion peptide-DOPA, anchoring peptide-RGD, and osteogenic-inducing peptide-BMP-2, as coatings on porous titanium alloy scaffolds printed on a 3D printer. In a rabbit model of bone defect, it was found that the implanted scaffolds with the bioactive coating stimulated osteointegration and exhibited mechanical stability [170]. However, the simple application of bioactive peptides onto metallic surfaces allows for only a small amount to be adsorbed, and due to the low affinity of the protein to metallic surfaces, rapid release and penetration into the systemic circulation are observed. To successfully stimulate osteoinduction, large amounts of rhBMP-2 (up to 1.50 mg/mL) are required [171]. However, the rapid release of high doses of bioactive peptides from the matrix increases the risk of complications, including ectopic bone formation, antibody formation against BMP, excessive bone resorption, and possibly the development of oncological diseases [172]. Therefore, biomimetic coatings saturated with BMP-2 have been used in studies with porous titanium. For this purpose, hydroxyapatite (HA) and synthetic and natural polymer films are used. These strategies are mainly aimed at maintaining an effective local concentration of BMP for a longer period [173]. One possible way to ensure the long-term presence of BMP at the healing site is the transfection of host cells at the site of injury with hBMP-2 DNA, resulting in their secretion of hBMP-2 protein at the healing site for many days [174,175,176]. The use of biomimetic coatings will undoubtedly increase biocompatibility and improve the osteointegration of porous titanium matrices, but for the successful application of these technologies, methods need to be developed that promote controlled local release of bioactive molecules, promoting proliferation of precursor cells (Figure 4) without significant systemic and local adverse effects.

### 4.4. Cell Colonization

For the purposes of bone tissue engineering, MSCs are widely used due to their ability to proliferate and undergo osteogenic differentiation [177]. Titanium possesses stable biocompatibility and, according to some studies, even promotes cell adhesion and proliferation [178] (Figure 3). In in vitro studies, titanium mesh membranes with square openings ranging from 25 µm to 75 µm have been shown to promote cell adhesion and proliferation [179]. Functionalizing titanium via the application of bioactive coatings, particularly derivatives of hydroxyapatite, significantly enhances MSC osteogenic differentiation and angiogenesis in human umbilical vein endothelial cells [180]. Endothelial microvascular network plays an important role in osteogenesis, bone regeneration, and bone tissue engineering. Endothelial progenitor cells (EPCs) have a high angiogenic and vasculogenic potential. Colonization of EPC matrices enhances their vascularization and formation of new bone tissue. Additionally, EPCs enhance osteogenic differentiation and osteogenesis of MSCs [181]. However, co-implantation of MSCs and EPCs did not enhance matrix vascularization, presumably because MSCs themselves can stimulate angiogenesis [182,183]. The use of precursor cells with osteogenic and vasculogenic potential for populating matrices used in orthopedics is a promising direction. However, the limited number of studies in this area and the conflicting data obtained do not allow for a definitive determination of the role and significance of cellular technologies in creating bioimplants for bone defect replacement.

### 4.5. Clinical Studies of Porous Ti-Based Materials

In a clinical study, a porous titanium interbody cage was used in patients undergoing anterior cervical discectomy to achieve interbody fusion. The titanium cages were characterized by high porosity (80%) and large pore size (700 microns) to facilitate osteointegration. The results showed that the clinical effectiveness of the titanium cages was not significantly different from that of traditionally used polyetheretherketone with (auto) graft. However, faster consolidation was observed [184]. A more recent similar study on posterior lumbar interbody fusion using polyetheretherketone cages also did not reveal significant differences between the porous titanium cages and polyetheretherketone. However, it was suggested that porous titanium cages might reduce vertebral body subsidence and accelerate intervertebral fusion [185].

To achieve fusion in patients undergoing anterior cervical discectomy, 3D-printed porous titanium and polyetheretherketone interbody cages with autograft were used as cervical implants. 3D-printed porous titanium cervical implants demonstrated significantly better clinical outcomes. Although there were no differences between the groups after 12 months, the titanium cages led to faster vertebral consolidation [184]. In a clinical study, 51 patients with primary osteoarthritis of the hip joint were randomized into two groups. In the experimental group, a porous titanium construct backside was implanted, while in the control group, patients were given a conventional porous coated titanium cup. When assessing periacetabular bone mineral density two years after surgery and implant fixation, no significant differences were observed between the two groups [186]. Similar data were obtained in another clinical study involving 248 patients with total hip arthroplasty. The authors compared the clinical and radiological outcomes between the conventional Stryker Trident HA cup and the high-porosity titanium cup. Both options showed good clinical results; however, the porous titanium led to a significantly higher rate of radiolucent lines around the cups, which was considered an indicator of possible cup loosening [187]. An algorithm for the development of an osteoconstructive porous Ti bioimplant is schematically presented in Figure 5. The described clinical studies indicate the promising potential of porous titanium scaffolds, which have comparable clinical effectiveness to standard materials and, unlike polyetheretherketone, do not require the additional use of autologous bone when used with cages. The increasing interest in 3D-printed porous titanium in recent years is evident by the growing number of clinical trials registered on the website www.clinicaltrials.gov (accessed on 10 November 2023).

## 5. Concluding Remarks

To sum up the notes above, there are still many unresolved issues in development of advanced bioactive porous Ti scaffolds despite the ever-growing number of the dedicated studies. There is still a number of current challenges such as appropriate computer-aided design and numerical simulation of mechanical performance and permeability of 3D-prinetd porous structures, proper adjustment of AM parameters, desired matching of designed and as-printed structures, residual powder removal, oxidation, the need for post-processing and so on in order to adjust AM technology for reliable printing of porous metallic scaffolds.

As to biomedical issues, the optimal dimensions and geometry of pores are still not yet unambiguously identified. On the one hand, increasing pore size from 50–150 μm to 500–700 μm can assist perfusion of tissue fluids and neoangiogenesis, which provides proliferation and differentiation of osteoblast precursors. On the other hand, one shall pass between Scylla and Charybdis to keep the required mechanical performance of the products with pore enlargement, while an explosive growth of MSCs without a proper differentiation would not facilitate fibrosis instead of neo-bone formation.

It is still unclear if bioactive coatings stimulating osteogenesis can be purposefully applied to porous Ti matrices. Most often used biomimetic apatite/hydroxyapatite coatings do not still have tried-and-true technology of their deposition, providing homogeneous and reliable fixation on the porous material. Application of BMPs as bioactive components of porous Ti structures requires dedicated separate studies due to the risk of systemic complications and ectopic bone formation. A separate task is related to development of special fastening elements for fixing innovative porous titanium scaffolds to the patient’s native bone. This may also imply designing new medical tools for clinical orthopedic oncology.

Problems related to the colonization of porous scaffolds with MSCs are still not fully resolved as well. Although it has been demonstrated that MSCs can successfully colonize Ti alloys, it is not clear in which direction their differentiation would develop in the pores: would they promote neoangiogenesis and osteogenesis or lead to the development of connective tissues and implant failure. Therefore, populating scaffolds with more differentiated cells, such as osteoblasts, is a more promising approach.

Clinical applications of porous implants are currently focused on the field of interbody cages and hip arthroplasty. Alongside that, orthopedic oncology and traumatology require replacing defects of cortical/pipe and pelvic bones, which is of vital importance. Here, a problem of fixation should be considered in the case of cellular scaffolds to prevent the failure of their porous structures. The implant design should envisage special solid inserts for fastening elements and predict their reaction to the overall performance.

The problem of trapped fine powders in porous structures inherited from the L/EB-PBF processing is also rarely addressed in the literature. Further, titanium release due to tribocorrosion and biodegradation can entail an undesired inflammatory response. Even higher strength, fully dense Ti implants can emit wear products such as micro- and nanoparticles, resulting in peri-implantitis and implant loss [188]. The associated trouble with the trapped air with the potential risk of air embolism should also be considered in clinical practice when loading the porous structures with pharmaceuticals.

Using porous structures as platforms for targeted drug delivery seems to be a very attractive feature of bioactive implants. Orthopedic oncology is especially interested in this function in order to prevent local recurrence after conditional radical resection of the affected bone area. Pharmaceuticals used can represent not only classic cytostatics inhibiting relapse of tumor, but also growth factors stimulating neoosteogenesis and vasculogenesis, and targeted immune drugs and cells such as activated lymphocytes (Cytokine-induced killer cell, CIK) and genetically modified lymphocytes—CAR-T cells. This area is also still poorly explored.

Thus, a comprehensive task of developing bioactive implants using AM technologies represents one of the most actual challenges in the fields of life and materials sciences. There are still a number of current challenges, such as residual powder removal, matching of designed and printed structures, oxidation, the need for post-processing, and so on, in order to adjust AM technology for reliable printing of porous structures. Thus, although AM porous titanium implants are being introduced into clinical practice, there is still a large number of systematic interdisciplinary in-depth studies to be performed to create functional and safe bioimplants using modern technologies including 3D printing.

## Figures and Tables

**Figure 1 biomimetics-08-00546-f001:**
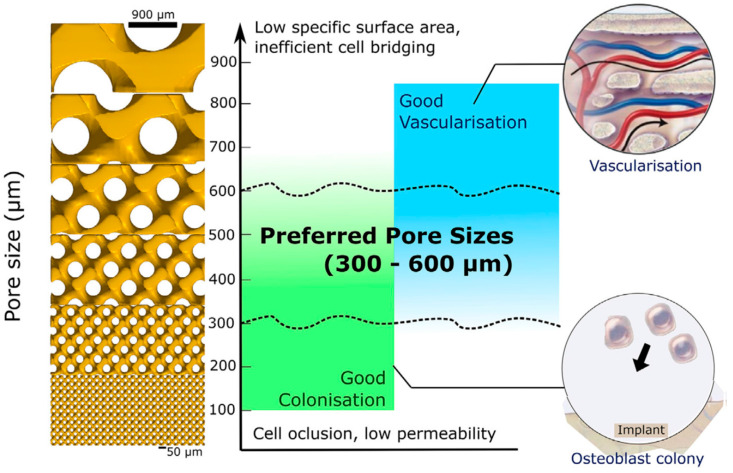
A tradeoff between bone colonization and vascularization in terms of pore size. The figure is reproduced from [34] under the terms of the CC BYNCND 4.0 license.

**Figure 2 biomimetics-08-00546-f002:**
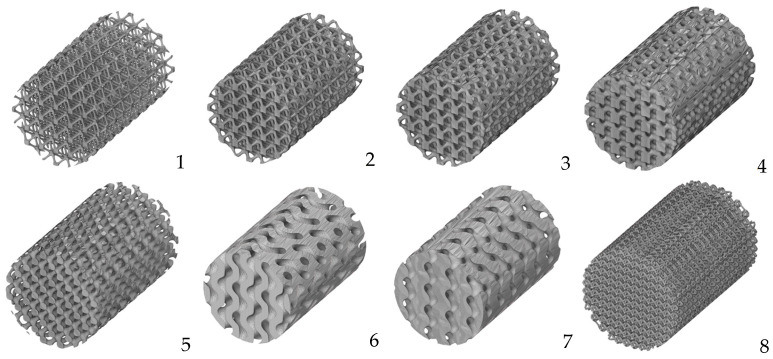
Models of a cylindrical sample built using TMPS models with different parameters featuring visualization of the different pore/wall ratio within IWP design (**1**–**4**) and different cell designs: Diamond (**5**) Gyroid (**6**,**7**) as well as strut-based octa-alike construction (**8**).

**Figure 3 biomimetics-08-00546-f003:**
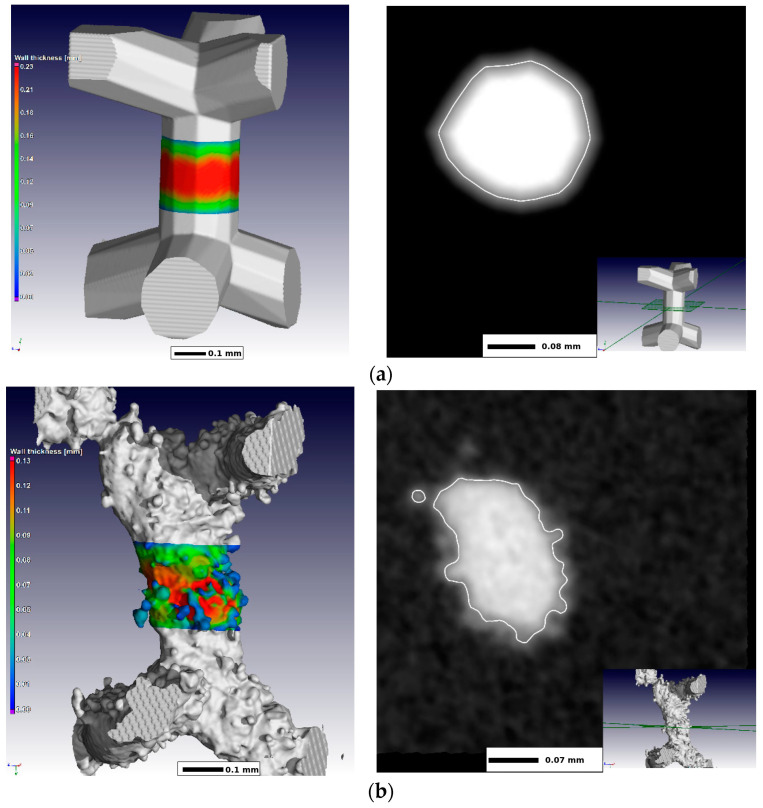
Micro CT images of (**a**) designed and (**b**) manufactured cell walls and their cross-sections. The figure is reproduced from [87] under the terms of the BB-CY license.

**Figure 4 biomimetics-08-00546-f004:**
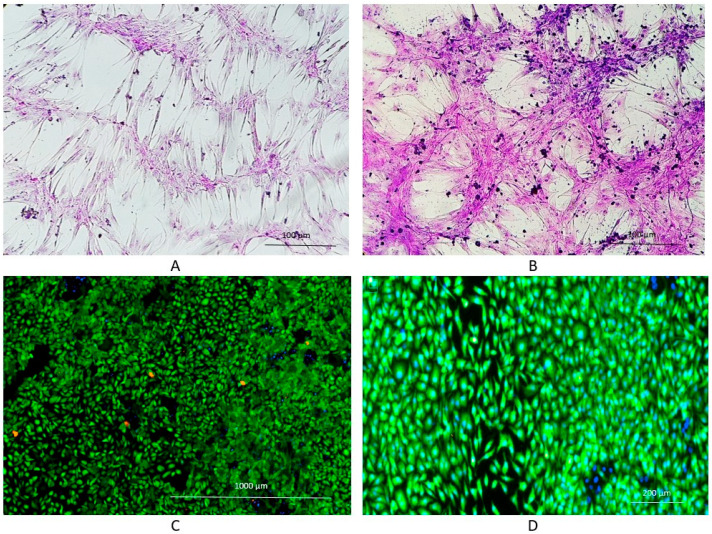
Proliferation of MSCs on plastic and titanium substrates in vitro. Proliferation of MSCs on plastic substrate original magnification 100 (**A**) and 400 (**B**), hematoxylin-eosin staining; Proliferation of MSCs on titanium substrate original magnification 100 (**C**) and 400 (**D**), green color—live cells Calcein AM staining, blue color—cell nuclei DAPI staining, red color—dead cells, propidium iodide staining (own unpublished data).

**Figure 5 biomimetics-08-00546-f005:**
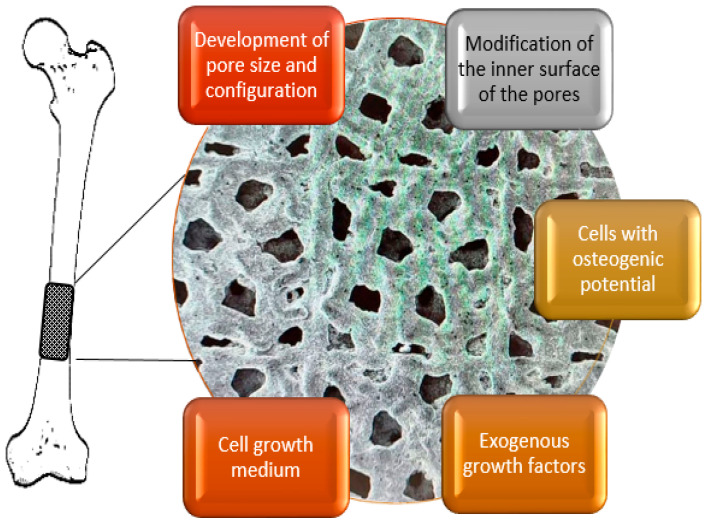
Algorithm for the development of an osteoconstructive porous Ti bioimplant.

**Table 1 biomimetics-08-00546-t001:** Model parameters.

Model	Expression
Diamond	cos X cos Y cos Z − sin X sin Y sin Z = c
IWP	2 (cos X cos Y +cos Y cos Z + cos Z cos X) − (cos 2X +cos 2Y +cos 2Z) = 0
Gyroid	sin Y cos X + sin Z cos Y + sin X cos Z = 0

## Data Availability

The data presented in this study are available on request from the corresponding author.

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
