# Peer review of "Development of Bioactive Scaffolds for Orthopedic Applications by Designing Additively Manufactured Titanium Porous Structures: A Critical Review"

_biomimetics, 2023, doi:10.3390/biomimetics8070546_

Round 1
Reviewer 1 Report (Previous Reviewer 1)
Comments and Suggestions for Authors
Ok for publication in the present form.
Comments on the Quality of English LanguageOk for publication in the present form.
Author Response
Thank you for your time and positive feedback on our work
Reviewer 2 Report (Previous Reviewer 2)
Comments and Suggestions for Authors
The manuscript ‘Development of bioactive scaffolds for orthopedic applications by designing additively manufactured titanium porous structures: a critical review’ has been substantially improved in relation to the first submission version. However, there are still issues which require revision.
1. The first phrase of Chapter 4 (l. 459-461) is taken from a reference, which describes its structure, and not the biomedical issues of porous scaffolds for BTE.
2. Chapter 4 starts referring to bioprinting, which has nothing to do with the rest of content. To my opinion, it should be omitted.
3. There is at least one wrong reference. Ref. 139 studies pore size in the range 188–390 μm, whereas the authors use this ref. to state that there are studies proposing as optimal pore size the range of 50-100 μm. This dictates the careful inspection of all references.
4. The conclusion of chapter 4.2 is, to this reviewer’s opinion, is not indisputable, but needs study.
5. The conclusion paragraph is a bit incoherent and needs rewriting.
6. There are phrases, throughout the manuscript, which are not very ‘academic’, e.g. l. 30 ‘the current troubles’, or have poor syntax, l.750 ‘summarizing … ‘.
Comments on the Quality of English LanguageContained in the previous text
Author Response
Please see the attachment

Reviewer 3 Report (Previous Reviewer 3)
Comments and Suggestions for Authors
Thank you to the authors for the revision of the paper, in my opinion it has improved substantially. I recommend publishing the paper as a good overview on the topic.
Just a few small issues/typos I found during reading:
Line 326: “structures”
Figure 3. It is hard to see what the actual dimensions are, thus, a readable scale bar would be welcome. Furthermore, the size of the powder used would be interesting to know.
Lines 466 and 469: I assume “bioink” is meant, not “bioinc”.
Line 750: “still” and “unresolved”
Line 756: “required”
Line 760: “applied”
Line 761: “deposition”
Round 2
Reviewer 2 Report (Previous Reviewer 2)
Comments and Suggestions for Authors
To this reviewer's opinion, the manuscript is substantially improved.
This manuscript is a resubmission of an earlier submission. The following is a list of the peer review reports and author responses from that submission.
Round 1
Reviewer 1 Report
Comments and Suggestions for Authors
The paper is interesting and deals with an interesting topic, both for industry and academia.
While the research is well structured and organized, the literature review seem to suffer. I would like to suggest to add following papers:
a very exhaustive review paper:
Alomar, Z., et al. Compressive behavior assessment of a newly developed circular cell-based lattice structure (2021) Materials and Design, 205, art. no. 109716. DOI: 10.1016/j.matdes.2021.109716
a paper describing new cell topologies:
Alomar, Z., et al. Compressive behavior assessment of a newly developed circular cell-based lattice structure (2021) Materials and Design, 205, art. no. 109716. DOI: 10.1016/j.matdes.2021.109716
A complete mechanical charactherization of the main widespread metals: Concli, F., et al. Experimental–numerical assessment of ductile failure of Additive Manufacturing selective laser melting reticular structures made of Al A357 (2021) Proceedings of the Institution of Mechanical Engineers, Part C: Journal of Mechanical Engineering Science, 235 (10), pp. 1909-1916. DOI: 10.1177/0954406219832333 Nalli, F., et al. Ductile damage assessment of Ti6Al4V, 17-4PH and AlSi10Mg for additive manufacturing (2021) Engineering Fracture Mechanics, 241, art. no. 107395, . DOI: 10.1016/j.engfracmech.2020.107395
ok
Reviewer 2 Report
Comments and Suggestions for Authors
none
Reviewer 3 Report
Comments and Suggestions for Authors
The authors claim to do a critical review on approaches to design special pore structures for bioactive scaffolds. Unfortunately, I have serious difficulties to get information on this topic from the paper. And I do not understand who the target readers should be.
The introduction (I see sections 1 and 2 as introduction) gives a quite good overview what the paper is about. However, in section 3 I started getting lost. This chapter is a very general description of SLM pros and cons readable in thousands of text books. If the target reader is someone who never heard of SLM, he might get a first impression, but for all others in my opinion it is meaningless and could be shortened to a one or two paragraphs. Furthermore, several times “sintering” is mentioned which is a completely different process compared to melting. The term is wrong in this context. On the other hand, today there is a number of interesting and powerful 3D printing techniques based on a final sintering process and suitable to create porous structures. They are not mentioned at all.
The next section, wrongly again numbered as “3” gives in 3.1 just descriptions of possible 3D-structures again you can find in text books. They are not used later. On the other side, the interesting modeling method Voronoi-tesselation used in section 4 is even not mentioned here.
The sentences are in most cases very general and not explained. For example lines 377 to 379: why are high-performance computers necessary? For what? It is not explained.
Section 3.3 (neural networks and AI): I do not understand what this section has to do with the topic of the paper. There is no related example, the text appears to be written for a typical funding application but not for a scientific journal. I recommend deleting it entirely.
Surprisingly, there is no transition from the AI part to the more technical one starting at line 480. In line 489 a review of existing model approaches of drug release is promised, but in the following not performed. There is one technique for 3D printing introduced (Photocuring 3D printing) but stated, this technique is not appropriate for metals (by the way, this is not true, there are companies doing this). The rest of this chapter is again written as a general text with some references but not as a “critical review”.
The part 4 about biomimetic scaffolds is the only section in my opinion which fulfills the promised intention of the paper. This is a quite nice overview on existing work and application. I would just recommend to integrate a few summarizing graphs or pictures. I think, using this chapter as the basis for a second version of the paper would make sense.
However, lines 863 to 867 are somewhat contradictory I think: first it is said, the porous part tends to possible loosening, than the statement is, that porous scaffold are promising. Maybe, a rewording should be done.
The Figure 4 is not mentioned at all in the text.
The conclusion is very general with no concrete examples of the “number of…”. And it summarizes not the paper, e.g. no FEM calculations of mechanical properties were actually shown .
To sum up, for me a red line of the paper is missing, details are missing, the text is much too general and figures and images are too spare. For me, a review comprises not only listing references, but taking information from these references, combining these informations to conclude new knowledge or hypotheses. This is not done in this paper.
Comments on the Quality of English LanguageOnly minor corrections necessary.